# AR-V7 exhibits non-canonical mechanisms of nuclear import and chromatin engagement in castrate-resistant prostate cancer

Seaho Kim[1†], CheukMan C Au[1†], Mohd Azrin Bin Jamalruddin[1†], Naira Essam Abou-Ghali[1], Eiman Mukhtar[1], Luigi Portella[1], Adeline Berger[2], Daniel Worroll[1], Prerna Vatsa[1], David S Rickman[2], David M Nanus[1,3], Paraskevi Giannakakou[1,3]*

[1]Department of Medicine, Weill Cornell Medical College, New York, United States; [2]Department of Pathology, Weill Cornell Medical College, New York, United States; [3]Meyer Cancer Center, Weill Cornell Medical College, New York, United States

*For correspondence:
pag2015@med.cornell.edu

†These authors contributed equally to this work

Competing interest: The authors declare that no competing interests exist.

**Abstract** Expression of the AR splice variant, androgen receptor variant 7 (AR-V7), in prostate cancer is correlated with poor patient survival and resistance to AR targeted therapies and taxanes. Currently, there is no specific inhibitor of AR-V7, while the molecular mechanisms regulating its biological function are not well elucidated. Here, we report that AR-V7 has unique biological features that functionally differentiate it from canonical AR-fl or from the second most prevalent variant, AR-v567. First, AR-V7 exhibits fast nuclear import kinetics via a pathway distinct from the nuclear localization signal dependent importin-α/β pathway used by AR-fl and AR-v567. We also show that the dimerization box domain, known to mediate AR dimerization and transactivation, is required for AR-V7 nuclear import but not for AR-fl. Once in the nucleus, AR-V7 is transcriptionally active, yet exhibits unusually high intranuclear mobility and transient chromatin interactions, unlike the stable chromatin association of liganded AR-fl. The high intranuclear mobility of AR-V7 together with its high transcriptional output, suggest a Hit-and-Run mode of transcription. Our findings reveal unique mechanisms regulating AR-V7 activity, offering the opportunity to develop selective therapeutic interventions.

## Editor's evaluation

This work performs a careful study of AR-V7, a splice variant of androgen receptor (AR) that lacks the androgen-binding domain, is constitutively active, and is typically expressed as prostate cancers become resistant to anti-androgen therapies. Clinically, there is intense interest in overcoming anti-androgen resistance, and part of this includes understanding differences between AR-V7 and AR, to be able to therapeutically target AR-V7. This manuscript provides a robust analysis of the regulation of nuclear import and the chromatin-binding features of AR-V7 versus AR. The work reveals that AR-V7 exhibits fast nuclear import kinetics in an NLS- and importin-α/β- independent manner, dependent on the dimerization (D-box) domain mediates AR-V7 nuclear import, revealing a new function for this domain versus its role in the full-length AR. The work also shows that AR-V7 employs an unconventional mode of transcription characterized by high intranuclear mobility, with transient and unstable chromatin interactions, likely reflecting a "Hit-and-Run" mechanism. This greatly enlarges the mechanistic understanding of AR-V7 function, and may help with developing new therapeutic agents.

## Introduction

Androgen receptor (AR) remains a critical therapeutic target in the treatment of metastatic castration-resistant prostate cancer (CRPC), due to overactive AR signaling (*Feldman and Feldman, 2001*). Next-generation AR inhibitors targeting either androgen biosynthesis (abiraterone acetate) or AR ligand binding (enzalutamide) have shown improved clinical outcomes including survival. However, these new therapies are not curative due to the development of resistance (*Watson et al., 2015*). Expression of active AR splice variants (AR-Vs) which re-activate AR transcriptional program in CRPC (*Antonarakis et al., 2016*; *Maughan and Antonarakis, 2015*) is one of the key drivers in disease progression and is believed to be one mechanism of resistance to abiraterone and enzalutamide. Structurally, the majority of AR-Vs lack the ligand-binding domain (LBD), which is the target of most AR-targeted therapies, and are constitutively active in the nucleus driving AR-signaling (*Uo et al., 2018*; *Watson et al., 2015*).

Among more than 20 alternatively spliced AR variants identified to date, AR-V7, which arises from cryptic exon inclusion, is the most prevalent variant in CRPC followed by the exon-skipping AR-v567. Expression of AR-V7 has been clinically associated with adverse patient outcomes including increase rates of metastases and inferior survival rates, and resistance to current standard of care treatment with abiraterone, enzalutamide, and taxane chemotherapy (*Antonarakis et al., 2014*; *Antonarakis et al., 2017*; *Guo et al., 2009*; *Hörnberg et al., 2011*; *Hu et al., 2009*; *Maughan and Antonarakis, 2015*; *Robinson et al., 2015* #80; *Rizzo et al., 2021*; *Robinson et al., 2015*; *Tagawa et al., 2019*). Together these data suggest that AR-V7 is a driver of CRPC progression and a desirable therapeutic target. However, the exact mechanism(s) underlying AR-V7 oncogenic functions are not well understood. Chromatin immunoprecipitation (ChIP), transcriptomic, and epigenetic studies have identified AR-V7 cistromes and target genes, both distinct and shared with AR-fl, as well as splicing factors that drive AR-V7 production, in an effort to elucidate potentially unique to AR-V7 regulatory mechanisms (*Cao et al., 2014*; *Hu et al., 2009*; *Li et al., 2013*; *Melnyk et al., 2020*; *Xu et al., 2015*).

Previously we showed that AR-fl binds microtubules (MTs) via the hinge domain and uses them as tracks for fast nuclear import (*Thadani-Mulero et al., 2014*; *Zhu et al., 2010*). Taxanes stabilize MTs and inhibit AR signaling by impairing AR-fl nuclear import and subsequent activation of target genes (*Antonarakis et al., 2017*; *Darshan et al., 2011*; *Thadani-Mulero et al., 2014*; *Zhu et al., 2010*). Similar to AR-fl, the hinge-containing AR-v567 binds MTs and is sensitive to taxane treatment. In contrast, the hinge-less AR-V7 does not bind MTs, conferring taxane resistance in xenografts and patients with CRPC (*Thadani-Mulero et al., 2014*).

In this study, we set out to investigate the mechanisms mediating AR-V7 nuclear import and its subnuclear biophysical properties in association with chromatin to identify unique, targetable biological features.

## Results

### AR-V7 exhibits fast nuclear import kinetics in a MT and importin-α/β independent pathway

Nuclear translocation is a pre-requisite for the transcriptional activity of AR-fl and all other nuclear hormone receptors. Nuclear import is mediated by a conserved bipartite nuclear localization signal (NLS) motif, which in AR-fl and AR-v567 is comprised of parts of exons 3 and 4. AR-V7 lacks exon 4, which is the second half of the canonical bipartite NLS, and although its cryptic exon 3 has been implicated in NLS reconstitution (*Chan et al., 2012*; *Guo et al., 2009*; *Hu et al., 2009*), the canonical NLS motif of AR-V7 is compromised. Yet, AR-V7 is constitutively localized to the nucleus in both cell lines and clinical samples (*Sun et al., 2010*; *Watson et al., 2010*), indicating efficient nuclear import. To measure basal nuclear import kinetics of the AR-Vs (AR-V7, AR-v567, and AR-fl), plasmids encoding each GFP-tagged AR were microinjected into the nuclei of AR-null PC3 cells and nuclear translocation kinetics was monitored by live-cell time-lapse confocal microscopy (*Figure 1A-B*). For each protein, we calculated the extent and rate of nuclear import by quantifying the % nuclear GFP-AR protein in single cells over time (*Figure 1C*). AR-fl remained largely in the cytoplasm under basal condition, exhibiting ~20% nuclear accumulation which remained steady over the duration of the experiment, indicating very low basal nuclear import kinetics. AR-v567 reached a maximum of ~50% by 90 min. In contrast, AR-V7 exhibited fast nuclear import kinetics, reaching ~50% nuclear accumulation within

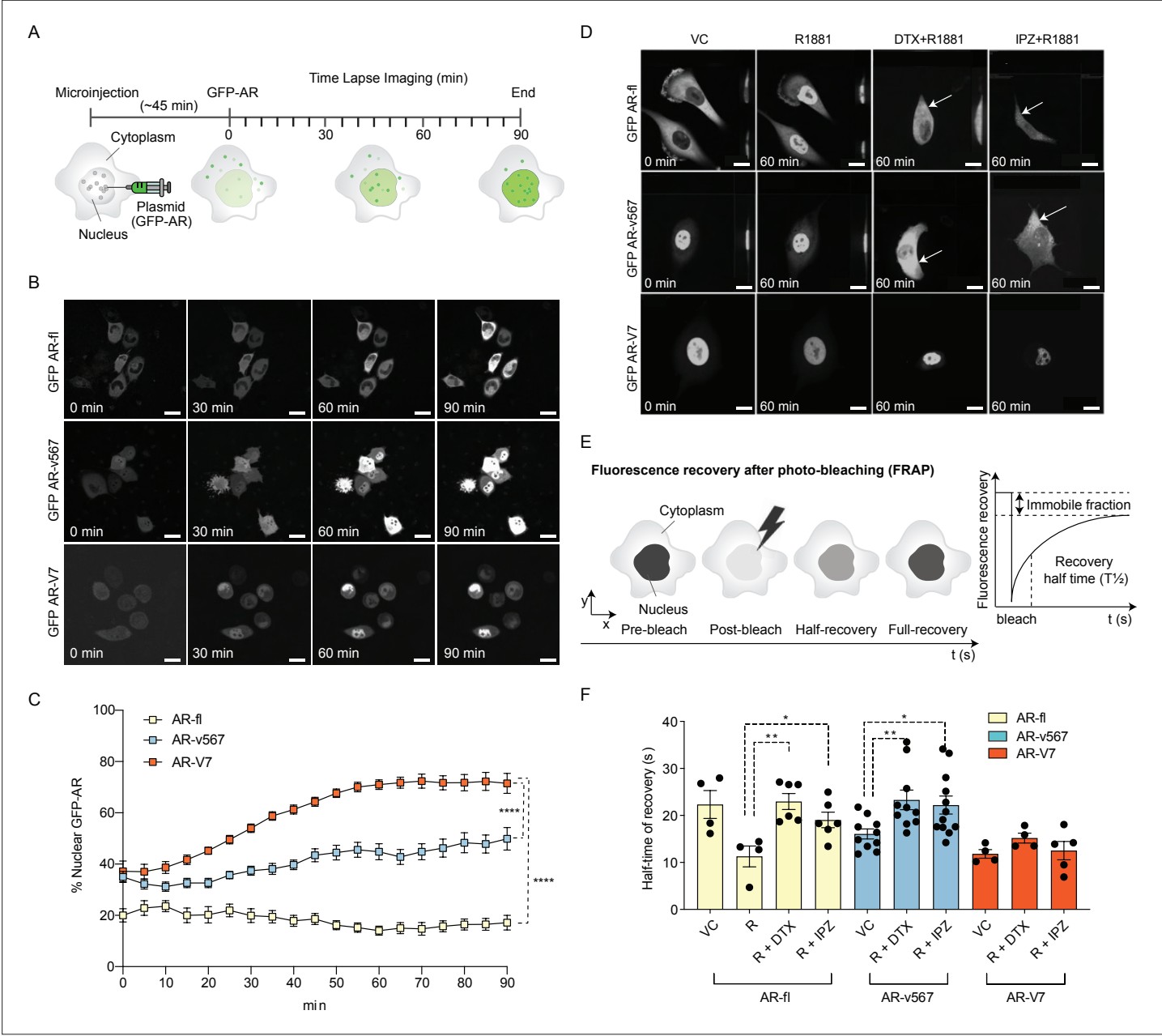

**Figure 1.** AR-V7 exhibits fast nuclear import kinetics independently of microtubules or the importin-α/β pathway, unlike AR-fl or AR-v567. (**A**) Experimental design. Plasmids encoding GFP-tagged AR-fl, AR-v567, or AR-V7 were microinjected into the nuclei of the AR-null PC3 cells. As soon as GFP-tagged proteins were detected in the cytoplasm (~45 min post micro-injection), nuclear translocation kinetics was monitored by live-cell time-lapse confocal microscopy at 5 min intervals for a total of 90 min. (**B**) Representative time-lapse images showing subcellular localization of each GFP-tagged AR protein. Scale bar, 10 µm. (**C**) Quantitation of % nuclear GFP-AR protein in single cells (n=3–10 cells/condition/time point). (**D**) Effect of MT and importin-β inhibitors on AR-fl, AR-v567, and AR-V7 nuclear localization. M12 prostate cancer cells stably expressing GFP-tagged AR-fl or AR-v567 or AR-V7 were treated as indicated and subjected to live-cell time-lapse imaging. R1881: synthetic androgen; DTX: docetaxel, MT-stabilizing drug; IPZ: importazole, importin-β inhibitor. Representative images are shown. Arrows point to cells with cytoplasmic GFP-AR-fl or GFP-AR-v567. Scale bar, 10 µm. (**E**) Schematic overview of Fluorescence Recovery After Photobleaching (FRAP) assay and its quantitative output. (**F**) Effect of MT and importin-β inhibitors on AR-fl, AR-v567, and AR-V7 nuclear translocation kinetics following FRAP. T1/2 times in s are shown for each respective protein (n=4–12 cells/condition). Data represent mean ± SEM, p value (*p<0.05, **p<0.01, ****$P<0.0001$) was obtained using unpaired two-tailed t-test. Experiments were repeated at least twice.

The online version of this article includes the following figure supplement(s) for figure 1:

**Figure supplement 1.** AR-V7 exhibits fast nuclear import kinetics independently of microtubules, actin, or the importin-α/β pathway.

**Figure supplement 2.** Dominant negative IPO11 does not abrogate the nuclear import of AR-fl or AR-V7.

the first 25 min and ~75% nuclear accumulation by 90 min (*Figure 1B-C* and *Figure 1—figure supplement 1A*). These data suggested that AR-V7 exhibited the highest basal nuclear import rates, likely via a more efficient nuclear import mechanism. It is established that canonical AR-fl utilizes the classical importin-α/β nuclear import mechanism where the importin-α binds to the NLS of AR protein followed by importin-β binding, forming a trimeric (cargo-NLS/importin-α/importin-β) complex in the cytoplasm which enters the nucleus through the nuclear pore complex (NPC) using the Ran-GTP (*Pemberton and Paschal, 2005*). To identify the mechanisms mediating AR-V7 nuclear import, we examined the involvement of the MT-transport system and the importin-α/β-Ran-GTP pathway (*Darshan et al., 2011*; *Jenster et al., 1993*; *Kaku et al., 2008*; *Thadani-Mulero et al., 2014*; *Zhou et al., 1994*; *Zhu et al., 2010*). We analyzed the translocation kinetics of each variant by live-cell time-lapse imaging using chemical probes that disrupt MTs (docetaxel [DTX]) or importin-β (Importazole [IPZ]) (*Figure 1D* and *Figure 1—figure supplement 1B–1D*). In agreement with our published data, AR-fl readily translocated to the nucleus upon addition of the synthetic AR ligand R1881, while perturbation of MTs with DTX, abrogated this effect (*Thadani-Mulero et al., 2014*). IPZ inhibited the R1881-induced AR-fl nuclear translocation, confirming the role of importin-β in the canonical AR-fl nuclear import pathway (*Kaku et al., 2008*), and that AR-v567 shares the same pathway of nuclear import with AR-fl (*Figure 1D* and *Figure 1—figure supplement 1C*). In contrast, neither DTX nor IPZ had an effect on AR-V7 nuclear localization, indicating that nuclear import of AR-V7 is both MT and importin-α/β independent (*Figure 1D* and *Figure 1—figure supplement 1D*).

To quantify nuclear translocation kinetics of AR proteins in response to treatment, we performed fluorescence recovery after photobleaching (FRAP) analysis in the M12 human metastatic prostate cancer cells stably expressing each GFP-tagged AR protein (*Thadani-Mulero et al., 2014*). The half-time of recovery (T½; defined as the time required for fluorescence intensity to reach 50% of its pre-bleach intensity) was then calculated and used as a readout of nuclear import dynamics (*Figure 1E*).

Treatment with R1881 accelerated AR-fl nuclear import by decreasing the T½ from 23 to 11 s while addition of DTX or IPZ significantly attenuated T½ to 23 and 19 s, respectively (*Figure 1F* and *Figure 1—figure supplement 1E*). The nuclear recovery of AR-v567, which retains the MT-binding hinge region, was also significantly impaired by DTX or IPZ treatment (T½ 23 and 22 s, respectively). In contrast, AR-V7 nuclear import kinetics was much faster than those of unliganded AR-fl (T½ 11 vs. 23 s) and was not affected by DTX or IPZ. To determine the involvement of the actin cytoskeleton in AR-V7 nuclear import, we treated PC3 cells microinjected with GFP-tagged AR plasmids with the actin-depolymerizing agent cytochalasin D (Cyto D) and identified that there was no effect on the nuclear import of AR-V7 nor in that of AR-fl or AR-v567 (*Figure 1—figure supplement 1F*).

## AR-V7 nuclear import requires active transport via the NPC and is partially dependent on Ran-GTP activity

Most nuclear import pathways involve the small GTPase Ran, which catalyzes the release of cargo protein from importin in the nucleus. As AR-V7 does not use importin-α/β pathway for nuclear import, we set out to determine whether it requires active transport via interaction with the nucleoporins, NPC components that mediate transport of proteins larger than 40 kDa. Thus, we incubated cells with wheat germ agglutinin (WGA), a well-established inhibitor of nucleoporin-mediated nuclear transport (*Finlay et al., 1987*; *Whitehurst et al., 2002*; *Yoneda et al., 1987*) and identified that WGA resulted in cytoplasmic sequestration of GFP-AR-V7 (*Figure 2—figure supplement 1A*), suggesting that interaction with nucleoporins is required for AR-V7 nuclear import. Next, to investigate whether AR-V7 depends on Ran-GTP for nuclear import, we quantified the nuclear fraction of GFP-tagged AR-fl, AR-v567, or AR-V7 proteins in the presence of the catalytic Ran-GTP mutant (mCherry-fused Ran Q69L; GTP hydrolysis deficient mutant). Our data identified that AR-fl and AR-v567 nuclear import was inhibited in the presence of the catalytic Ran-GTP mutant (*Figure 2A*); while AR-V7 nuclear import was partially inhibited, as evidenced by AR-V7 localization in both nucleus and cytoplasm (*Figure 2A*; solid arrows). Quantitation of the percent nuclear signal of each AR protein identified significant decrease in nuclear localization for all proteins in the presence of the catalytic mutant RanQ69L (*Figure 2B*). Similar results were observed when HEK293T cells were transiently co-transfected with GFP-AR-fl, GFP-AR-v567, or GFP-AR-V7 and mCherry-tagged Ran Q69L (*Figure 2—figure supplement 1B and C*). In addition, to confirm these results in cells expressing endogenous AR-fl, we generated C4-2 cells stably expressing inducible GFP-AR-V7 and examined its subcellular localization, following induction with doxycycline,

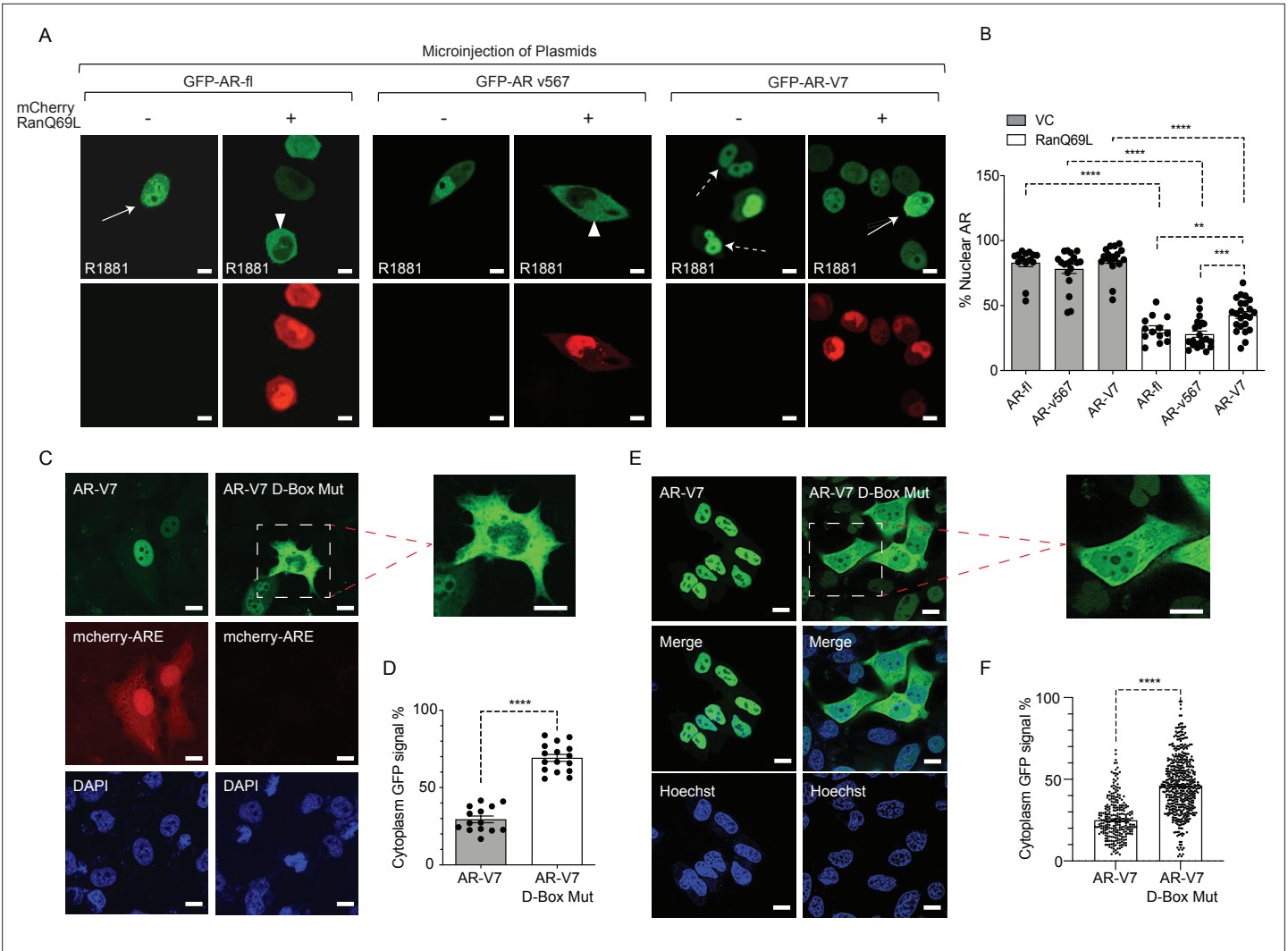

**Figure 2.** Inactivation of Ran-GTP-mediated nuclear transport affects differentially the subcellular localization of AR-variants. (**A, B**) Plasmids encoding GFP-tagged AR-fl, AR-v567, or AR-V7 were co-microinjected with plasmid encoding the catalytic mutant mCherry-tagged RanQ69L, into the nuclei of the AR-null PC3 cells. Cells expressing both tagged proteins were subjected to live-cell time-lapse imaging. Cells were treated with R1881 (10 nM) to induce AR-fl nuclear import and was kept present with the variants so that all conditions were the same. Expression of mCherry-tagged RanQ69L affected differentially the subcellular localization of each GFP-tagged AR proteins. Solid arrow: cell with both cytoplasmic and nuclear AR proteins; arrowheads: cytoplasmic AR proteins; dashed arrow: nuclear AR proteins. % Nuclear AR across conditions is graphically displayed in B (n>10 per condition). (**C–F**) AR-V7 nuclear import is impaired upon mutation of the dimerization box domain (D-box). PC3 cells stably expressing ARE-mCherry reporter were transfected with GFP-AR-V7 or GFP-AR-V7 D-box mutant (A596T and S597T). Representative images are shown (inset displays higher magnification of the indicated cell) and quantitative results are graphically displayed in (**D**) (n>10 cells per condition). (**E, F**) C4-2 cells stably expressing inducible GFP-AR-V7 or GFP-AR-V7 D-Box mutant were used to quantify subcellular AR-V7 localization following doxycycline induction. Representative images are shown (inset displays higher magnification of the indicated cell) and quantitative results are graphically displayed (**F**). ( n> 200 cells per condition). Data represent mean ± SEM, p value (**p<0.01, ***p<0.001, ****p<0.0001) was obtained using unpaired two-tailed t-test. Scale bar, 10 μm. Experiments were repeated at least twice.

The online version of this article includes the following figure supplement(s) for figure 2:

**Figure supplement 1.** AR-V7 nuclear import requires active transport via the nuclear pore complex is dependent on Ran-GTP activity and is impaired upon mutation of the dimerization box domain (D-box).

in the presence of the catalytic mutant RanQ69L. We observed, using live-cell imaging, enhanced cytoplasmic localization of AR-V7 in cells co-expressing the Ran mutant, consistent with our earlier findings (*Figure 2—figure supplement 1D*). Interestingly, when the mutant Ran was introduced into the 22RV1 cells endogenously expressing both AR-fl and AR-V7, expression of AR-V7 appeared to be downregulated in the presence of the mutant Ran (*Figure 2—figure supplement 1E*).

## AR-V7 nuclear import is impaired upon mutation of the dimerization box domain (D-box)

Androgen-regulated gene expression requires AR-fl receptor dimerization in the nucleus mediated by the zinc finger (D-box) domain, prior to DNA binding. AR-V7 transcriptional activity has been shown to depend on the D-box domain which mediates AR-V7 homodimerization in the nucleus (*Centenera et al., 2008*; *Xu et al., 2015*). To determine the potential impact of D-box domain on AR-V7 nuclear localization and transcriptional activity, we generated a construct encoding GFP-AR-V7 containing two functionally inactivating D-box mutations (A596T and S597T) and transiently transfected PC3 cells stably expressing the ARE-mCherry reporter (*Azeem et al., 2017*). Single-cell analysis showed nuclear GFP-AR-V7 was transcriptionally active as evidenced by concurrent mCherry-ARE expression. In contrast, the GFP-AR-V7-D-box mutant was both transcriptionally inactive and enriched in the cytoplasm compared to wild-type GFP-AR-V7 (29% in cytoplasmic AR-V7 vs. 71% in AR-V7-D-box mutant; p<0.0001) (*Figure 2C-D*). Interestingly, the same D-box mutations had no effect on AR-fl nuclear localization (*Figure 2—figure supplement 1F*) consistent with the role of D-box on AR-fl homodimerization in the nucleus. To expand these results to additional cell lines, we generated C4-2 cells harboring endogenous AR-fl, to stably express inducible GFP-AR-V7-D-Box mutant and quantified its subcellular localization by live-cell imaging. Our results indicated significantly enhanced cytoplasmic localization of the AR-V7-D-box mutant in comparison to its inducible wild-type counterpart (*Figure 2E-F*). Taken together, these results indicate that AR-V7 nuclear import partially requires an intact D-box domain, identifying a novel, variant-specific function for this conserved domain.

## AR variants drive ligand-independent fractional nuclear translocation of AR-fl with no evidence of heterodimerization

In CRPC, AR-fl is often co-expressed with AR variants (AR-V) in patient tumors and it has been suggested that active AR signaling in castrate conditions is partially due to AR-V heterodimerization with AR-fl resulting in its nuclear translocation in the absence of ligand (*Cao et al., 2014*; *Xu et al., 2015*). To determine the effect of AR-V7 on AR-fl nuclear localization, we microinjected mCherry-AR-fl with GFP-AR-V7 in PC3 cells and quantified AR-fl nuclear accumulation across conditions (*Figure 3A-B*). Under basal conditions, we observed low AR-fl nuclear localization which was increased by twofold in the presence of AR-V7 in the absence of ligand (12% vs. 24%, p<0.001). Similar results were observed in PC3 cells and C4-2 cells following transient transfection with mCherry-AR-fl and GFP-AR-V7 (*Figure 3—figure supplement 1A–1B*). Interestingly, a similar pattern of enhanced AR-fl nuclear localization in the absence of ligand, was observed in the presence of GFP-AR-v567 (*Figure 3—figure supplement 1C*), suggesting that this interaction is not unique to AR-V7 and may apply to additional nuclear AR-variants. As AR-V7 is the most prevalent AR-V expressed in patient tumors, we sought to determine next whether enhanced nuclear localization of AR-fl in the presence of AR-V7 occurs in an androgen sensitive model of prostate cancer. To that end, we transfected LNCaP cells with mCherry-AR-fl and/or GFP-AR-V7. The percentage of nuclear AR-fl was numerically higher in the presence versus in the absence of AR-V7 co-expression (31% vs. 25%) (*Figure 3C-D*).

Mechanistically, we sought to determine whether AR-V7 might form a heterodimer with unliganded AR-fl in the cytoplasm driving the latter into the nucleus. Treatment with IPZ showed that when AR-V7 and AR-fl were co-expressed in the same cell, IPZ inhibited the nuclear localization of AR-fl only; while it had no effect on AR-V7 (*Figure 3E*), consistent with our earlier results (*Figure 1*). These results do not support cytoplasmic AR-fl/AR-V7 heterodimerization, in agreement with recently published reports (*Chen et al., 2018*). Next, we sought to determine whether nuclear AR-V7/AR-fl heterodimerization might underlie the enhanced nuclear retention of AR-fl. As the D-box domain was previously shown to be partially involved in AR-fl/AR-V7 heterodimerization (*Roggero et al., 2021*), we generated C4-2 cells stably expressing inducible GFP- AR-V7 or GFP-AR-V7- D-Box mutant, in addition to GFP-AR-V7 previously generated, and transfected them with mCherry-AR-fl. Our results show no difference in % nuclear AR-fl when co-expressed with either wild-type or D-box mutant AR-V7 (*Figure 3F-G*). Taken together, these results argue against the presence of physical interaction between AR-fl and AR-V7 in the cytoplasm or nucleus as the underlying mechanism of enhanced unliganded nuclear AR-fl.

To investigate whether AR-V7 transcriptional output mediates the fractional AR-fl nuclear translocation, we introduced the DNA-binding domain (DBD) mutation (A573D) known to abrogate the transcriptional activity of canonical AR, into AR-V7 (AR-V7-A573D). We found that the A573D mutation

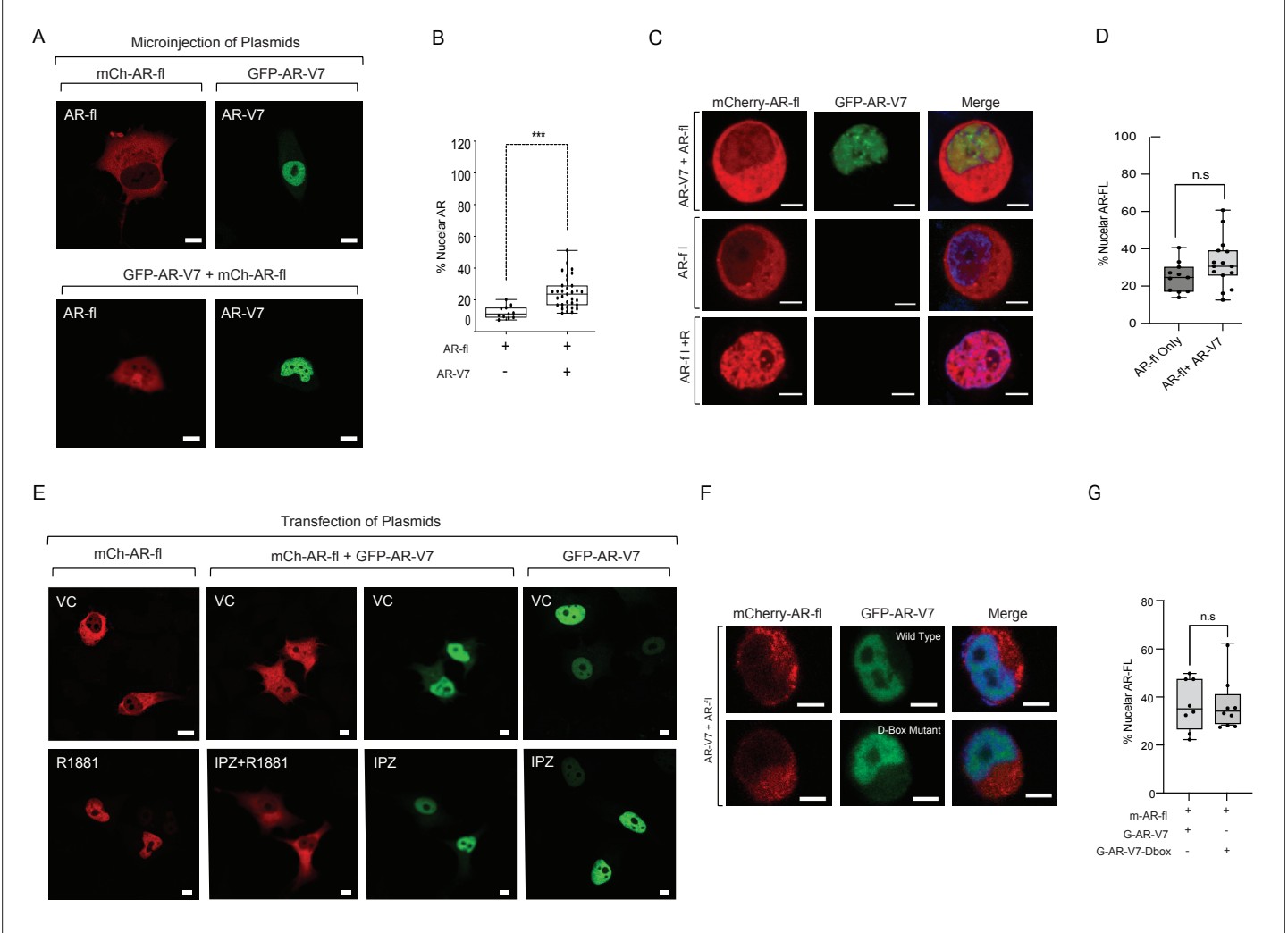

**Figure 3.** AR-V7 drives nuclear translocation of AR-fl in the absence of ligand. (**A, B**) Plasmids encoding mCherry-AR-fl or GFP-AR-V7 were micro-injected in PC3 cells. Representative microscopic images (scale bar, 10 μm) and % nuclear AR is shown. Data represent a box plot with n>10cells per condition, p value (***p<0.001) was obtained using unpaired two-tailed t-test. (**C, D**) LNCaP cells were transfected with mCherry-ARfl and/or GFP-AR-V7, and cells were treated 10 nM R1881 as indicated (scale bar, 5 μm). Data quantification is shown as a box plot and was obtained using unpaired t-test with n≥10 cells per condition. (**E**) PC3 cells were transfected with mCherry-AR-fl, or GFP-AR-V7, and cells were treated 10 nM R1881 or 50 μM Importazole, as indicated. Representative confocal microscopy images are shown with arrows pointing to nuclear AR-fl across conditions. Scale bar, 10 μm. Experiments were repeated at least twice. (**F, G**) C4-2 cells stably expressing inducible GFP-AR-V7 or GFP-AR-V7 D-Box mutant were induced by doxycycline and transfected with mCherry-AR-fl (scale bar, 5 μm). Representative images are shown, data quantification is shown as a box plot, and obtained using a Mann-Whitney test with n≥ 8 per condition.

The online version of this article includes the following figure supplement(s) for figure 3:

**Figure supplement 1.** AR variants enhance nuclear localization of AR-fl in the absence of ligand.

abrogated AR-V7 transcriptional activity, but not its nuclear localization, compared to unaltered AR-V7 (***Figure 4A-C***). This is evidenced by the decrease in ARE-mCherry reporter activity and the decreased expression of the endogenous AR-V7 target gene, FKBP5. When the transcriptionally inactive AR-V7-A537D mutant was co-expressed with AR-fl, we observed a similar increase in nuclear AR-fl in the presence of mutant or unaltered AR-V7 as compared to baseline levels (31% vs. 39 vs. 19%, respectively, p<0.0001) (***Figure 4D-E***).

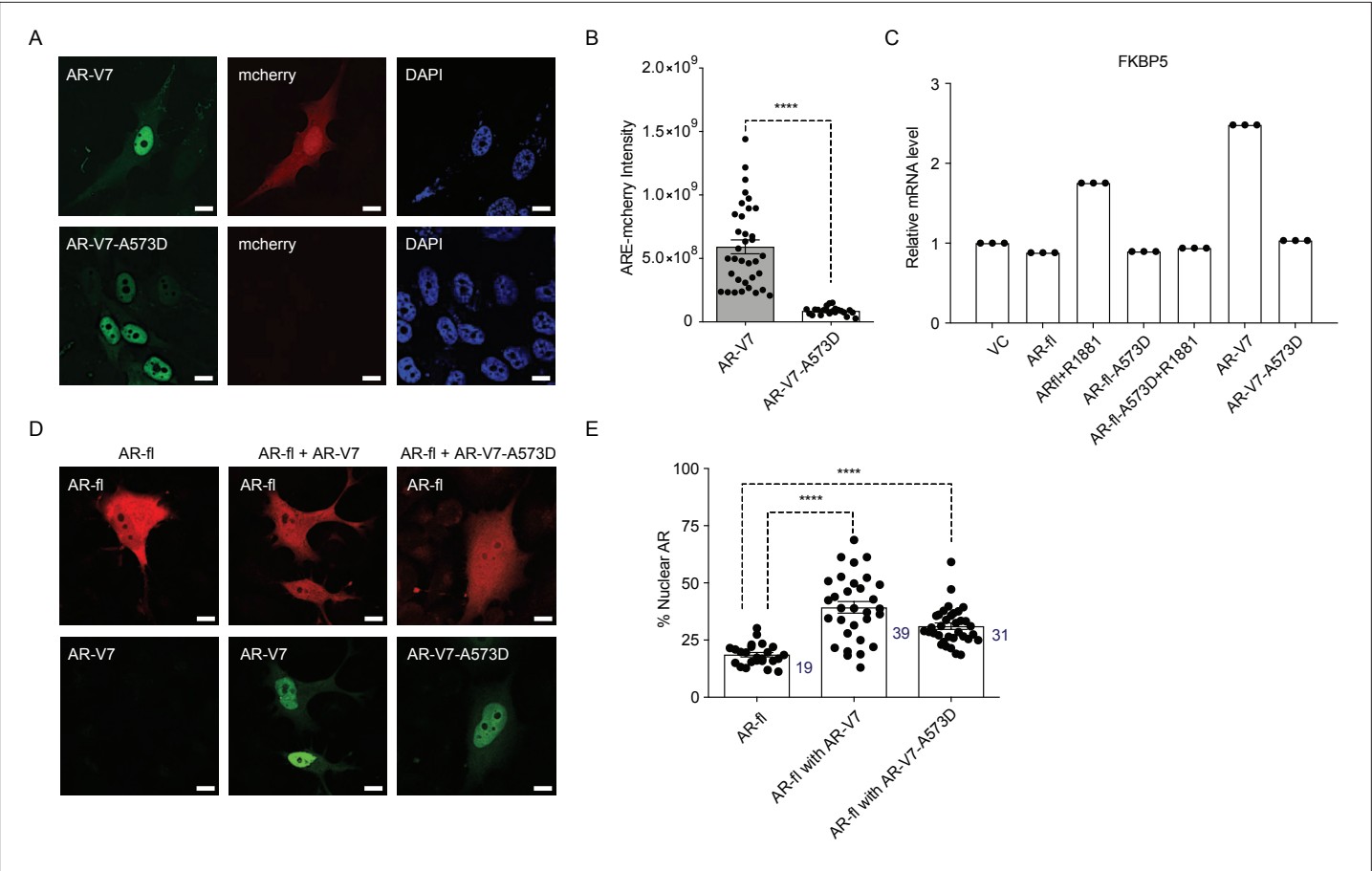

**Figure 4.** DBD mutation abrogates AR-V7 transcriptional activity (**A, B**) PC3 cells stably expressing ARE-mCherry reporter were transfected with the indicated plasmids for 48 hr and the expression of GFP protein with concomitant mCherry protein was analyzed by confocal imaging. Representative images and quantitative results are shown (n>10 cells per condition). Scale bar, 10 μm. (**C**) RT-qPCR for the endogenous FKBP5 mRNA was quantified in PC3 cells after transfection of indicated plasmids. Data with AR-fl A573D are included as a control (n=3). (**D**) PC3 cells were transfected with mCherry-AR-fl or GFP-AR-V7 or GFP-AR-V7-A573D and imaged by confocal microscopy (scale bar, 10 μm) and (**E**). % Nuclear AR protein was quantified (n>23 cells per condition). Data represent mean ± SEM, p value (****p<0.0001) was obtained using unpaired two-tailed t-test. Experiments were repeated at least twice.

## AR-V7 exhibits high subnuclear mobility kinetics and short chromatin residence time

It is well established that agonist-bound AR-fl is transcriptionally active and relatively immobile in the nucleus; while antagonist-bound AR-fl is highly mobile and transcriptionally inactive (*Farla et al., 2004*; *Farla et al., 2005*). AR-V7, on the other hand, is transcriptionally active in the absence of agonist-binding, however, the exact mechanism underlying its nuclear activity is not known. Thus, we investigated the exchange rate of AR-V7 with chromatin using FRAP and confocal live-cell imaging in cells transfected with GFP-tagged AR-V7 or AR-fl as a control. FRAP analysis of AR-fl identified slow fluorescence recovery of ligand-bound nuclear AR was significantly delayed compared to unliganded AR (T½~8 s vs. 3 s, respectively, p<0.0001) (*Figure 5A-C*), suggesting prolonged chromatin residence time of ligand-bound AR, in agreement with reports on AR and other nuclear hormone receptors including the glucocorticoid, estrogen, and progesterone receptors (*Farla et al., 2004*; *Farla et al., 2005*; *Klokk et al., 2007*). In contrast, the fluorescence recovery of nuclear GFP-AR-V7 was very fast compared to R1881-bound AR-fl (T½~4 s and 8 s, respectively, p<0.0001) (*Figure 5A-C*), indicating that AR-V7 exhibits short chromatin residence time, despite being transcriptionally active.

To better understand the kinetics of subnuclear mobility of each protein, we generated photoconvertible mEos4b-AR-fl or mEos4b-AR-V7 and followed them by live-cell imaging (*Paez-Segala et al., 2015*). Photo-conversion of AR-fl in a small sub-nuclear region, led to a change in the

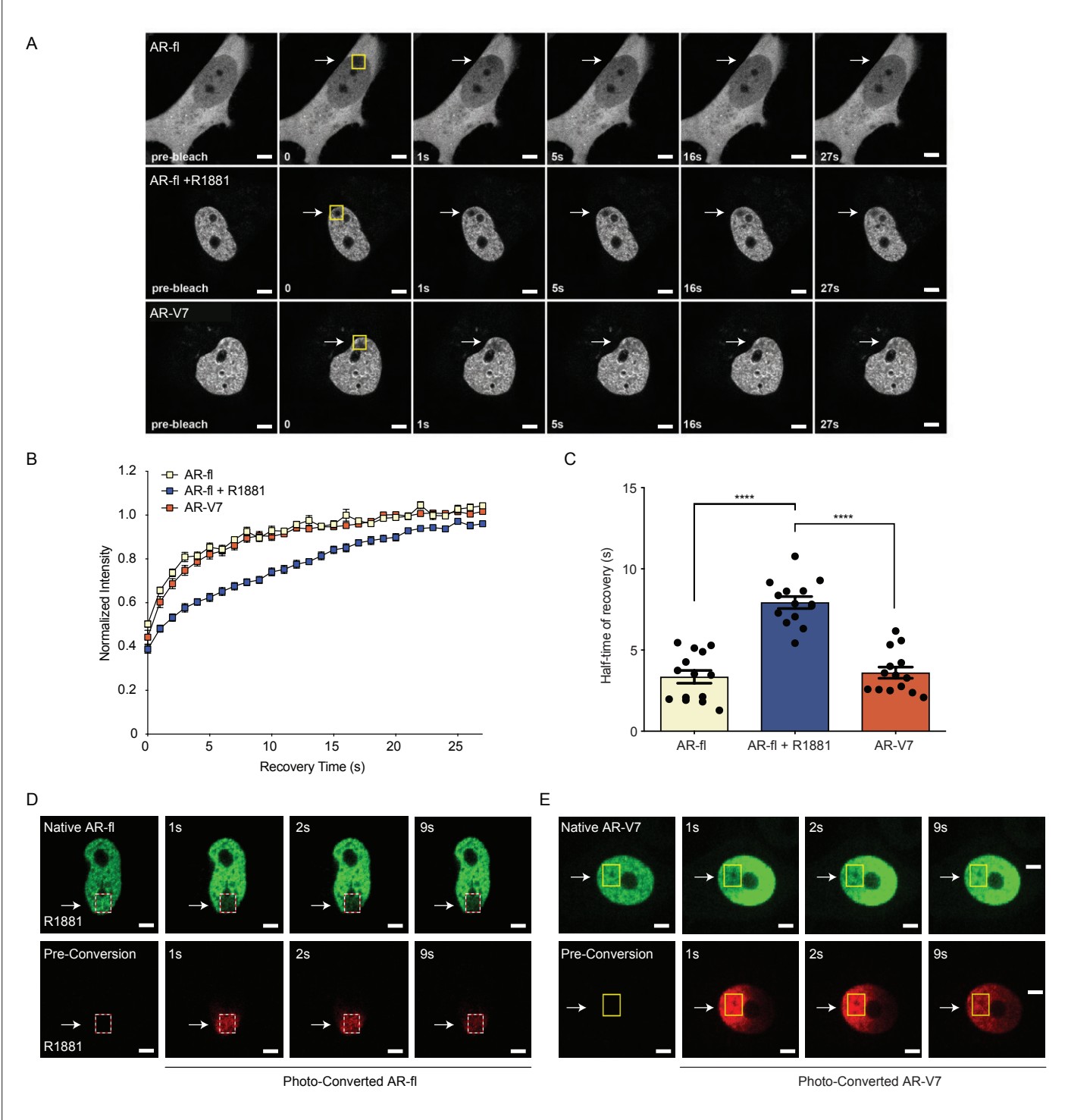

**Figure 5.** AR-V7 exhibits high intranuclear mobility compared with liganded AR-fl. (**A**) FRAP was performed in PC3 cells transiently expressing GFP-AR-fl (in the absence or presence of 10 nM R1881) or GFP-AR-V7. FRAP was monitored at 1 s intervals. Representative images of cells at select time points are shown. (**B**) Kinetics of proteins recovery after photobleaching at 1 s intervals are graphically displayed, n=14. (**C**) Graphic display of half-time of recovery (T1/2) in seconds (**s**) for each condition, n=14. (**D–E**) FRAP was performed in PC3 cells transiently expressing photoconvertible mEos4b-AR-fl or mEos4b-AR-V7 protein. Cells were imaged at 1 s intervals to monitor fluorescence recovery of the non-converted proteins (Green) and nuclear distribution of photo-converted proteins (Red). Scale bar, 10 μm. Data represent mean ± SEM, p value (****p<0.0001) was obtained using unpaired two-tailed t-test. Experiments were repeated at least twice.

The online version of this article includes the following figure supplement(s) for figure 5:

**Figure supplement 1.** AR-V7 intranuclear mobility is not affected by co-expression of ligand-bound AR-fl.

fluorophore from green (unconverted) to red (converted), which was monitored by time-lapse imaging at 1 s intervals (*Figure 5D*, dotted box). Our data identified that AR-fl remained within the confines of the photoconverted area, without fluorescence recovery by unconverted AR (green) suggesting stable and prolonged chromatin binding (*Figure 5D*). On the other hand, photo-converted AR-V7, in <1 s started moving outside the photoconverted area (*Figure 5E*, yellow box), repopulating the entire nucleus in less than 9 s. This high intranuclear mobility of AR-V7 was accompanied by rapid fluorescence recovery of unconverted-AR-V7 (green) (*Figure 5E*). These data reveal for the first time a sharp distinction between the chromatin exchange rates and intranuclear mobility of AR-V7 and canonical ligand-bound AR-fl, despite their overlapping cistromes. To investigate any potential nuclear interactions of both proteins when co-expressed in the same cell, we co-microinjected mCherry-AR-fl and GFP-AR-V7 in the nuclei of PC3 cells and monitored their respective recovery after photobleaching. We observed a similar pattern of intranuclear kinetics for each protein, when co-expressed in the same sub-nuclear area, as the kinetics observed when each protein was expressed alone with fast AR-V7 fluorescence recovered vversus slow recovery of ligand-bound AR-fl (*Figure 5—figure supplement 1A–1C*).

## DNA binding mutation abrogates AR-V7 transactivation and accelerates nuclear mobility kinetics

To resolve the conundrum between the high nuclear mobility of AR-V7 and its high transcriptional activity, we introduced the A573D DBD mutation into AR-V7 or AR-fl expression plasmids and introduced them into PC3 cells. We detected a significant increase in the mobility of the mutant AR-fl-A573D compared to AR-fl, in the presence of ligand, with T ½ of 4 s versus 17 s, respectively, p<0.0001 (*Figure 6A* and *Figure 6—figure supplement 1*). No difference was observed in the absence of ligand, suggesting that immobilization of the ligand-bound AR-fl is mediated by theDBD of AR-fl. Surprisingly, we found that the already high intranuclear mobility of AR-V7 was further accelerated by the DBD mutation with recovery T ½ for AR-V7-A573D at 2 s vversus 4 s for AR-V7, p<0.0001 (*Figure 6B*). These data suggested that DNA-binding mediates the transient chromatin interactions exhibited by AR-V7. Next, we examined the effect of the DBD mutation on the transcriptional output of AR-fl and AR-V7, using ARE-mCherry expression as a transcriptional readout in single cells, or target gene mRNA expression by RT-qPCR in cell populations. Our results revealed that the A573D mutation abrogated transcriptional activity in both, ligand-bound AR-fl and AR-V7, as evidenced by the significant decrease in ARE-mCherry expression (*Figure 6C-D*). Taken together, these data couple DNA binding with nuclear mobility kinetics and AR-V7 transactivation.

To investigate whether the high intranuclear mobility of AR-V7 is due to its reduced occupancy rate on target AREs on chromatin, we performed ChIP for AR-V7 or AR-fl in 22Rv1 cells with endogenous expression of both proteins. ChIP-qPCR data showed that R1881 increased the occupancy of AR-fl on the enhancer regions of PSA and FKBP5 (*Figure 6E*). In contrast, AR-V7 showed very low occupancy rate on PSA and FKBP5 enhancer (~0.01% input) compared to ligand-bound AR-fl occupancy (6–12% Input). (*Figure 6F*). To corroborate these data, we performed subcellular fractionation of C4.2: GFP-AR-V7 cells expressing endogenous AR-fl and doxycycline-inducible AR-V7 (*Figure 6G*). As expected, R1881 enhanced both the nuclear AR-fl (NE) and chromatin-bound AR-fl (CB) fractions (*Figure 6G*). In contrast, minimal if any AR-V7 was detected in the chromatin-bound fraction, consistent with its low chromatin occupancy rate and high subnuclear mobility kinetics.

## Discussion

Inhibition of androgen receptor signaling remains the cornerstone of contemporary therapeutic strategies for patients with metastatic CRPC. Reactivation of AR signaling is a hallmark of CRPC, largely mediated by the nuclear activity of the AR splice variant AR-V7 (*Antonarakis et al., 2014*; *Antonarakis et al., 2017*; *Guo et al., 2009*; *Hörnberg et al., 2011*; *Hu et al., 2009*; *Maughan and Antonarakis, 2015*).

In this study, we identify that AR-V7 nuclear import does not use the canonical NLS-dependent importin-α/β pathway, in contrast to AR-fl and AR-v567. Earlier findings suggested that the C-terminal cryptic exon 3 (CE3) domain of AR-V7 might mediate its nuclear import due to it similarity with the second bipartite NLS of AR-fl (*Chan et al., 2012*). The same study, also showed that a synthetic

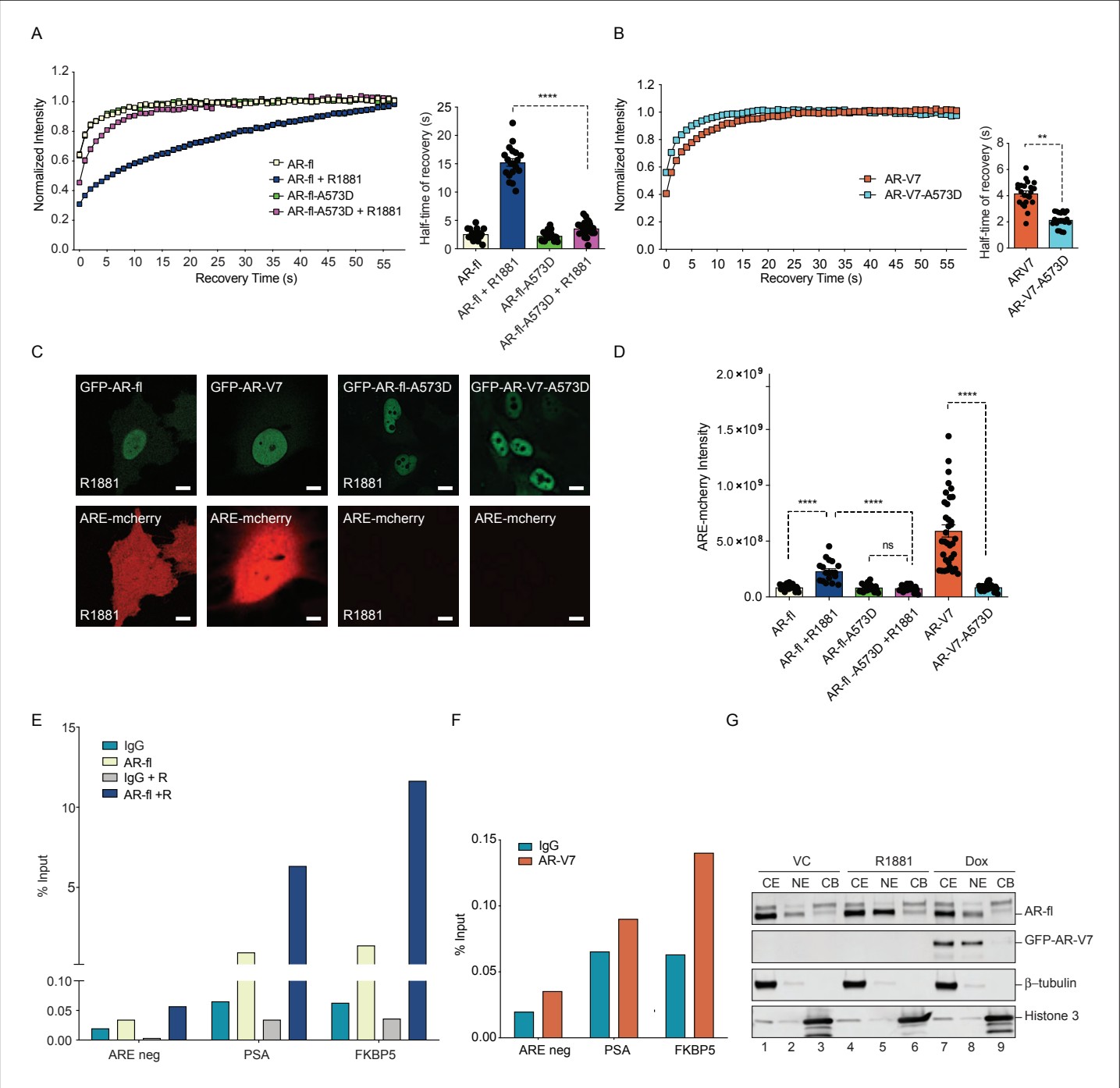

**Figure 6.** DNA-binding mutation increases the intranuclear mobility and abrogates the transcriptional activity of AR-fl and AR-V7. (**A, B**) FRAP was performed in PC3 cells transiently expressing GFP-AR-fl or GFP-AR-V7 or their respective DBD mutants (A573D). Kinetics of protein recovery after photobleaching are graphically displayed and their half-time of recovery, is obtained for each condition, n>14. (**C**) PC3 cells stably expressing ARE-mCherry reporter were transfected with indicated plasmids, in the presence or absence of ligand (10 nM R1881). Representative images of each condition are shown. (**D**) quantitation of mCherry fluorescence intensity in single cells (n>17). (**E, F**) The binding of (**E**) AR-fl or (**F**) AR-V7 on the enhancer of PSA or FBKP5 in 22RV1 was analyzed by ChIP-QPCR assay. Cells in charcoal stripped media were treated with vehicle or 10 nM R1881 for 24 hr. (**G**) Immunoblot for AR-fl and AR-V7 following subcellular fractionation CE, cytosolic extract; NE, nuclear extract, CB, chromatin-bound nuclear extract. Histone H3 and β-tubulin were used as controls for the fractionation. Data represent mean ± SEM, p value (**p<0.01, ****p<0.0001) was obtained using unpaired two-tailed t-test. Experiments were repeated at least twice.

The online version of this article includes the following source data and figure supplement(s) for figure 6:

*Figure 6 continued on next page*

*Figure 6 continued*

**Source data 1.** DNA-binding mutation increases the intranuclear mobility and abrogates the transcriptional activity of AR-fl and AR-V7.

**Figure supplement 1.** DBD mutation increases the intranuclear mobility of liganded-AR-fl and AR-V7.

truncated AR-V7 lacking the CE3 domain, did not bind importin-β, yet was transcriptionally active in the nucleus. These data not only show the CE3 motif is dispensable for nuclear import, but they are also consistent with our results where pharmacologic inhibition of importin-β, has no effect on AR-V7 nuclear localization confirming an NLS-independent mechanism of AR-V7 nuclear entry. The molecular weight of AR-V7 protein together with its cytoplasmic sequestration upon nucleoporin inhibition suggest that AR-V7 requires active transport via the NPC, likely using nuclear transporters known to recognize proteins without a classical bipartite NLS (*Figure 2—figure supplement 1A*). Importin 11 (IPO11), is such an alternative nuclear transport receptor, recently shown to mediate PTEN nuclear import (*Chen et al., 2018*). Using a catalytic mutant IPO11 we observed no effect on AR-V7 nuclear import, suggesting that other nuclear transport receptors may be involved. In ongoing work beyond the scope of this manuscript, we are systematically testing the impact of each importer on AR-V7 nuclear localization. Alternative mechanisms of nuclear import may rely on the recognition of post-translational modifications that mediate AR-V7 recognition by another beta-like nuclear importer (*Putker et al., 2013*).

Surprisingly, we found that the same inactivating mutations introduced in the D-box domain (A596T and S597T), identified in patients with androgen insensitivity syndrome (*Centenera et al., 2008*), had a different effect on AR-fl compared to AR-V7. It is well established that the D-box domain is important for AR dimerization in the nucleus,which occurs prior to DNA binding, and that mutations in this domain impair nuclear AR dimerization and activation of target genes (*van Royen et al., 2012*). Using bimolecular fluorescence complementation assay, (*Xu et al., 2015*) showed that D-box mutations in AR-V7 inhibited AR-V7 homodimerization in the nucleus. In addition, our results (*Figure 2*) revealed that D-box mutant AR-V7 is transcriptionally inactive because it is sequestered in the cytoplasm, suggesting that the D-box domain may be important for the nuclear translocation of AR-V7, which is distinct from its function in AR-fl.

Analysis of single PC3 cells co-expressing tagged AR-FL and AR-V7, showed significant increase in AR-fl nuclear localization, in the absence of ligand (*Figure 3C* and *Figure 3—figure supplement 1*) in agreement with recent reports (*Cao et al., 2014*). Taken together, these results suggested a potential physical interaction between the two proteins in the cytoplasm and a shared mechanism of nuclear import. To test this hypothesis, we used IPZ to inhibit importin-α/βmediated AR-fl nuclear import and observed no effect on AR-V7, implying that AR-V7 and AR-fl use independent nuclear import pathways and that likely there is no physical interaction between the two proteins in the cytoplasm (*Figure 3C*). Along these lines, previous studies showed that AR-V7 does not co-precipitate with AR-fl and that they form a physical complex in the nucleus (*Chen et al., 2018*; *Guo et al., 2009*; *Roggero et al., 2021*). Interestingly, when the transcriptionally inactive AR-V7-A537D mutant was co-expressed with AR-fl, we observed a similar fractional increase in AR-fl nuclear translocation, in the absence of ligand (*Figure 4D-E*) implicating non-transcriptional mechanisms. It should be noted that the mere localization of AR-fl in the nucleus in castrate conditions does not imply AR signaling activation. Taken together our own findings and published results, we posit that there is no evidence that AR-V7 mediated fractional increase in nuclear AR-fl contributes to the overall activity of AR-V7 in driving castrate-independent growth in PC.

This study also revealed unique nuclear biology of AR-V7, distinct from that of ligand-bound AR-fl, suggesting a distinct mode of transcriptional action. The chromatin binding dynamics of nuclear hormone receptors (glucocorticoid, progesterone, estrogen, and androgen) have been closely correlated with their respective transcriptional output and affinity to their ligand (*Farla et al., 2004*; *Farla et al., 2005*; *Fletcher et al., 2000*; *Klokk et al., 2007*; *McNally et al., 2000*; *Stenoien et al., 2001*). It is well established that AR-V7 is constitutively active in the nucleus having largely overlapping cistromes and target genes with canonical AR-fl (*Cato et al., 2019*). Herein, using live-cell imaging and photo-conversion to quantify subnuclear dynamics of AR proteins, we show that ligand-bound AR-fl is transcriptionally active and exhibits low intranuclear mobility, prolonged residence time on chromatin. and high occupancy rates on promoter AREs (*Figures 5 and 6*). Conversely, AR-V7 is transcriptionally active, yet exhibits unusually high intranuclear mobility and transient chromatin interactions with low

occupancy rates. The high intranuclear mobility of AR-V7 together with its high transcriptional output, suggest a Hit-and-Run model of transcription, where a transcription factor (TF) transiently binds a DNA sequence to regulate target genes (the 'hit'), and before vacating the site (the 'run') recruits secondary TFs which form stable complex at the regulatory site that sustain a stable long-term effect (*Charoensawan et al., 2015*). Interestingly, the Hit-and-Run is often applied to transcriptional repressors, where gene silencing does not necessarily require continuous TF residence on chromatin (*Shah et al., 2019*). Importantly, a recent study showed that AR-V7 functions as a transcriptional repressor in CRPC, preferentially binding several co-repressors compared to AR-fl, likely due to differences in H3K27 acetylation (*Cato et al., 2019*). This mode of transcription is likely to be promoted by structural differences relative to AR-fl and post-translational modifications specific to AR-V7. It has also been shown that the chromatin residence of nuclear receptors, such as estrogen receptor, can be slowed in the presence of antagonist (*Guan et al., 2019*), and the effect of N terminal domain AR inhibitors on the intranuclear mobility of AR-V7 remains to be determined. Finally, the reported diversity of AR-V7 regulated transcriptomes across patients with CRPC, which likely results from the cell-context specific AR-V7 cistromes (*Chen et al., 2018*) is compatible with a Hit-and-Run mode of transcription that allows for fast adaptation to environmental cues. The role of these intrinsic differences between AR isoforms in promoting their unique modes of transcriptional action remains to be investigated. The precise machinery regulating the rapid nuclear import of AR-V7 and its therapeutic relevance will be a subject of future investigation.

## Materials and methods

### Cell lines

PC3, LNCaP, C4-2, HEK-293T, and 22Rv1 cell lines were obtained from the ATCC. We generated the C4-2 cell line (tet-on GFP-AR-V7 or GFP-AR-V7-D-Box mutant), which stably expresses tetracycline-inducible GFP-AR-V7 or GFP-AR-V7-DBox mutant by infecting the lentiviral construct (detailed information in Plasmid Constructions section). The stable M12 cell lines expressing GFP-tagged AR-fl, AR-v567, or AR-V7 were described previously (*Thadani-Mulero et al., 2014*). Mycoplasma detection of all cell lines were tested, and negative results were observed. Authenticated cell lines were used within 6 months of purchase from the ATCC.

### Antibodies and reagents

Primary antibodies were used: rabbit polyclonal anti-AR-N-terminal (AR-N-21), rabbit monoclonal anti-AR-V7, rabbit monoclonal anti-AR-C-terminal, rabbit polyclonal anti-actin, rabbit polyclonal anti-beta tubulin, rabbit polyclonal anti-Histone H3, rabbit polyclonal anti-GFP, mouse monoclonal anti-mCherry (1C51). DTX (Taxotere), IPZ, and cytochalasin D were obtained from Sigma-Aldrich. WGA Alexa Fluor 594 Conjugate (W11262) were purchased from molecular probes. See Key resources table for antibodies used in this study.

### Plasmid constructions

The following plasmids pmCherry-AR-fl, pEGFP-C1-AR-fl, pEGFP-C1-AR-v567, and pEGFP-AR-V7 (*Thadani-Mulero et al., 2014*) were used for transfection or microinjection into the cell nuclei. The DBD mutant at A573D of AR-fl and AR-V7 in the pEGFP-C1 backbone were generated by site-directed mutagenesis using the Q5 Site-Directed Mutagenesis Kit (New England BioLabs). The dimerization box (D-box) mutant at A596T/S597T of AR-V7 in the pEGFP-C1 backbone was also generated by the same mutagenesis approach.

The photoconvertible mEos4b-C1 backbone (Addgene plasmid #54812) was a gift from Dr. Michael Davidson (*Paez-Segala et al., 2015*). We subcloned AR-fl, AR-v567, or AR-V7 in mEos4b-C1 to generate N-terminally tagged-photoconvertible AR constructs (mEos4b-AR-fl, mEos4b-AR-v567, and mEos4b-AR-V7) used for live-cell imaging. Doxycycline inducible GFP-tagged AR-V7 or GFP-tagged AR-V7-DBox mutant was generated by subcloning into the lentiviral pCW57.1 tet-on vector (a gift from Dr. David Root, Addgene Plasmid #41393) using Gateway cloning (Invitrogen). The construct containing mCherry-tagged GTP-hydrolysis defective Ran mutant, pmCherry-C1-RanQ69L (Addgene plasmid #30309), was a gift from Dr. Jay Brenman (*Kazgan et al., 2010*) and used for live-cell imaging. ARE-reporter vector CS-GS241B-mCHER-LV152 with mCherry fluorescent reporter signal was a gift

from Dr. Karl-Henning Kalland (*Azeem et al., 2017*). Using this vector, we generated lentiviral particles to infect PC3 cells. Stable PC3 cells harboring CS-GS241B-mCHER-LV152 were generated by hygromycin (500 µg/ml) selection and used for ARE-mCherry reporter assay.

## Transient transfections of plasmid

90,000 cells were plated on coverslips and transfected with or without plasmid (refer to Plasmid constructions section) using FuGENE HD (Promega) or AMAXA Nucleofector R Kit (Lonza), according to the supplier's instructions. Transfected cells were fixed or imaged live within 24–48 hr after transfection and 4–8 hr of doxycycline or R1881 treatment. Cells were analyzed using Hoechst nuclear stain, confocal microscopy, and ImageJ.

## Live-cell imaging, FRAP, and photo-conversion analysis

Live-cell imaging was carried out on cells either microinjected or transfected with the plasmids described above. Cells were grown on No. 1.5 coverglass mounted on 35 mm MatTek dish and cultured in RPMI 1640 supplemented with 5% charcoal-stripped FBS, 25 mM HEPES, 2 mM sodium pyruvate, and 2 ml L-glutamine. Live-cell imaging and FRAP were carried out on a Zeiss LSM 700 confocal microscope equipped with an on-stage live-cell chamber (Tokai Hit, Shizuoka, Japan). Photo bleaching in the region of interest was carried out with the 405 nm laser at maximum power for three iterations. A single z-section was imaged before and at time intervals. The normalized intensity of region of interests and the half-time of recovery required for the fluorescence intensity to reach 50% of its pre-bleach intensity ($T_{1/2}$) were obtained using Zeiss Zen software. Photo-conversion imaging analysis was performed in PC3 cells plated on 35 mm MatTek dish after transient transfection with photoconvertible AR constructs described above. PC3 cells transfected with mEos4b-AR-fl was further incubated with 10 nM R1881 for 1 hr before the photo-conversion. Briefly, region of interest at nuclei of cells was photo-converted by applying 405 nm laser (60% power for 400 ms) using the Mosaic system (Andor, Oxford Instruments, UK) equipped with spinning disk confocal microscope (Zeiss/Perkin-Elmer) at Bio-imaging resource center at the Rockefeller University. The time-lapse images were captured in two different channels (for green 491 nm laser, 525–50 nm filter; for red 561 nm laser, and 620–60 nm filter) before and after photo-conversion and images acquisition was performed with MetaMorph software.

## ARE-mCherry reporter assay

PC3 cells stably expressing the ARE-reporter vector with mCherry fluorescent reporter signal was plated on 35 mm MatTek dish. GFP-tagged AR-fl or AR-V7 was microinjected in the nuclei of cells. The synthetic androgen R1881 was added to the cells microinjected with GFP-AR-fl construct. After overnight incubation, GFP-AR and ARE-reporter mCherry signal were imaged using Zeiss scanning confocal microscope.

## Quantitative real-time PCR

For relative quantitation of AR target genes, quantitative real-time PCR was performed on 100 ng input RNA using Power SYBR green RNA-to-Ct 1 step kit (Applied Biosystems) and primers specific for PSA (F: 5'-ACGCTGGACAGGGGGCAAAAG, R: GGGCAGGGCACATGGTTCACT), FKBP5 (F: 5'-GCGGAGAGTGACGGAGTC, R: 5'-TGGGGCTTTCTTCATTGTTC), and ACTIN (F: 5'-CCTCCCTGG AGAAGAGCTA, R: 5'-CCAGACAGCACTGTATTGG). Relative quantitation was used to determine fold change in expression levels by the comparative Ct method.

## Chromatin Immunoprecipitation

22RV1 cells were trypsinized and crosslinked in 1% formaldehyde media for 10 min at room temperature and quenched for 8 min using 125 mM glycine. Nuclear extracts were collected and sonicated for 10 min to obtain 300 bp chromatin fragments (Diagenode Bioruptor Pico). Equal volumes of sheared chromatin were immunoprecipitated with rabbit AR-V7 antibody (RevMab 31-1109-00), rabbit AR antibody (Abcam 52615), or rabbit IgG control (Santa Cruz Biotechnology). After extensive washing, crosslinking was reversed, and DNA fragments were purified using Macherey-Nagel kit (740609). Q-PCR amplification was performed using the ABI 7500 fast system (Fast SYBR Green 4385612 Applied Biosystems) and the relative standard curve method in a 96-well format. Primers

used: FKBP5e: F-GGT TCC TGG GCA GGA GTA AG; R-AAC GTG GAT CCC ACA CTC TC; PSAe: F-TGG GAC AAC TTG CAA ACC TG; R-GAT CCA GGC TTG CTT ACT GT; AREneg: F-GCT GAT TCA ATT ACC TCC CAG AA; R-AGT TTG GGA CAG ACG GGA AA. The input chromatin for each sample was analyzed at four concentrations (serial dilutions) to generate a standard curve per primer pair and per 96-well plate. The sheared chromatin was diluted 1/6 before being used for Q-PCR. All reactions were run in triplicate.

## Subcellular fractionation and Western blot analysis

C4.2 cell line expressing tet-inducible GFP-AR-V7 was treated with either 10 nM R1881 for AR nuclear translocation or 1 µg/ml doxycycline for GFP-AR-V7 induction. Subcellular fractionation of cells was performed using the subcellular protein fractionation kit (Thermo Fisher Scientific) and each extract was subjected to immunoblotting with indicated antibodies.

## Statistical analyses

The Student's two-tailed t-test was used to determine the mean differences between two groups. $p < 0.05$ is considered significant. Data were presented as mean ± SEM.

## Acknowledgements

The authors thank Dr. Alison North for assistance with the photo-conversion study and Dr. Ved P Sharma for assistance with imaging data analysis relating to study of AR-V7-Dbox mutant from the Bio-Imaging Resource Center at Rockefeller University. The authors are grateful to Dr. Karl-Henning Kalland and Dr. Waqas Azeem from University of Bergen, Norway (ARE-mCherry reporter), Dr. David Root from Broad Institute (pCW57.1 tet-on vector, Addgene Plasmid #41393), Dr. Jay Brenman from University of North Carolina at Chapel Hill (pmCherry-C1-RanQ69L, Addgene plasmid #30309) for provision of reagents and valuable experimental advice. The authors acknowledge Dr. Urko D Castillo and William G Stone IV for assistance with plasmid cloning work. This work was supported by grants from the US NIH T32 CA203702 (S Kim), US NIH T32 CA062948 (C C Au), R01CA137020 (P Gianna-kakou), R21CA216800 (P Giannakakou), R01CA228512 (P Giannakakou); R01CA179100 (P Gianna-kakou and D S Rickman); and from the Department of Defense W81XWH-17-1-0162 (A Berger).

# Additional information

## Funding

| Funder | Grant reference number | Author |
| --- | --- | --- |
| National Cancer Institute | NIH T32 CA203702 | Seaho Kim |
| National Cancer Institute | NIH T32 CA062948 | CheukMan C Au |
| National Cancer Institute | R01CA137020 | Paraskevi Giannakakou |
| National Cancer Institute | R21CA216800 | Paraskevi Giannakakou |
| National Cancer Institute | R01CA228512 | Paraskevi Giannakakou |
| National Cancer Institute | R01CA179100 | David S Rickman Paraskevi Giannakakou |
| U.S. Department of Defense | W81XWH-17-1-0162 | Adeline Berger |
| National Cancer Institute | F32CA220988 | Eiman Mukhtar |

The funders had no role in study design, data collection and interpretation, or the decision to submit the work for publication.

## Author contributions

Seaho Kim, CheukMan C Au, Mohd Azrin Bin Jamalruddin, Conceptualization, Data curation, Formal analysis, Funding acquisition, Validation, Investigation, Methodology, Writing - original draft, Writing

- review and editing; Naira Essam Abou-Ghali, Investigation, Methodology, Writing - review and editing; Eiman Mukhtar, Luigi Portella, Adeline Berger, Investigation, Methodology; Daniel Worroll, Resources, Investigation, Methodology; Prerna Vatsa, Resources, Supervision, Investigation, Methodology, Writing - review and editing; David S Rickman, Conceptualization, Resources, Formal analysis, Supervision, Funding acquisition, Methodology, Writing - original draft, Project administration, Writing - review and editing; David M Nanus, Resources, Supervision, Investigation, Methodology; Paraskevi Giannakakou, Conceptualization, Resources, Formal analysis, Supervision, Funding acquisition, Investigation, Methodology, Writing - original draft, Project administration, Writing - review and editing

### Author ORCIDs
CheukMan C Au ⓘ http://orcid.org/0000-0001-8289-7365
Naira Essam Abou-Ghali ⓘ http://orcid.org/0000-0002-7143-9577
Daniel Worroll ⓘ http://orcid.org/0000-0001-7351-676X
Paraskevi Giannakakou ⓘ http://orcid.org/0000-0001-7378-262X

### Decision letter and Author response
Decision letter https://doi.org/10.7554/eLife.73396.sa1
Author response https://doi.org/10.7554/eLife.73396.sa2

---

## Additional files

### Supplementary files
• Transparent reporting form

### Data availability
All data generated or analysed during this study are included in the manuscript. Source data files have been provided for figure 6.

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

# Appendix 1

## Appendix 1—key resources table

| Reagent type (species) or resource | Designation | Source or reference | Identifiers | Additional information |
|---|---|---|---|---|
| Cell line (*Homo sapiens*) | PC3 | ATCC | RRID: CVCL_4885 | |
| Cell line (*H. sapiens*) | LNCaP | ATCC | RRID: CVCL_0395 | |
| Cell line (*H. sapiens*) | C4-2 | ATCC | RRID: CVCL_4782 | |
| Cell line (*H. sapiens*) | HEK-293T | ATCC | RRID: CVCL_0063 | |
| Cell line (*H. sapiens*) | 22rv1 | ATCC | RRID: CVCL_1045 | |
| Cell line (*H. sapiens*) | M12 | ATCC | RRID: CVCL_4860 | |
| Antibody | (Rabbit polyclonal) anti-AR-N-terminal (AR-N-21) | This paper | | Custom antibody produced by the Giannakakou Lab (1:500) |
| Antibody | (Rabbit monoclonal) anti-AR-V7 | RevMab | Cat #: 31-1109-00 RRID: AB_2716436 | (1:500) |
| Antibody | (Rabbit monoclonal) anti-AR-C-terminal | Abcam | Cat #: ab52615 RRID: AB_867653 | (1:1000) |
| Antibody | (Rabbit polyclonal) anti-actin | Sigma-Aldrich | Cat #: A2066 RRID: AB_476693 | (1:500) |
| Antibody | (Rabbit polyclonal) anti-beta tubulin | Abcam | Cat #: ab6046 RRID: AB_2210370 | (1:500) |
| Antibody | (Rabbit polyclonal) anti-Histone H3 | Abcam | Cat #: ab1791 RRID: AB_302613 | (1:1000) |
| Antibody | (Rabbit polyclonal) anti-GFP | Novus Biologicals | Cat #: NB600-303 RRID: AB_10001300 | (1:500) |
| Antibody | (Mouse monoclonal) anti-mCherry 1C51 | Novus Biologicals | Cat #: NBP1-96752 RRID: AB_11034849 | (1:500) |
| Antibody | Wheat Germ Agglutinin (WGA) - Alexa Fluor 594 Conjugate | Molecular Probes | Cat #: W11262 RRID: AB_2334867 | 5 µg/ml |
| Antibody | (Goat anti-Rabbit) IgG (H+L) Cross-Adsorbed Secondary Antibody, Alexa Fluor 488 | Thermo Fisher Scientific | Cat#: A-11008 | (1:10,000) |
| Antibody | (Goat anti-Mouse) IgG (H+L) Cross-Adsorbed Secondary Antibody, Alexa Fluor 568 | Thermo Fisher Scientific | Cat#: A-11004 | (1:10,000) |
| Chemical compound, drug | Docetaxel (Taxotere) | Sigma-Aldrich | Cat #: 01885 | 1 µM |
| Chemical compound, drug | Importazole | Sigma-Aldrich | Cat #: SML0341 | 50 µM |
| Chemical compound, drug | Cytochalasin D | Sigma-Aldrich | Cat #: C8273 | 1 µg/ml |
| Chemical compound, drug | Hygromycin | Santa Cruz Biotechnology | Cat #: sc-29067 | (1:1000) |
| Chemical compound, drug | Doxycycline | Enzo Life Sciences | Cat #: ALX380273G005 | 1 µg/ml |

*Appendix 1 Continued on next page*

*Appendix 1 Continued*

| Reagent type (species) or resource | Designation | Source or reference | Identifiers | Additional information |
|---|---|---|---|---|
| Chemical compound, drug | R1881 | Sigma-Aldrich | Cat #: R0908 | 10 nM |
| Chemical compound, drug | Puromycin Dihydrochloride | Thermo Fisher Scientific | Cat #: A1113803 | 1 µg/ml |
| Chemical compound, drug | Hoechst 33342, Trihydrochloride, Trihydrate | Thermo Fisher Scientific | Cat #: H3570 | 10 µg/ml |
| Transfected construct | pmCherry-AR-fl (PC3) | This paper | | Produced by Giannakakou Lab |
| Transfected construct | pEGFP-C1-AR-fl (PC3, M12) | This paper, *Thadani-Mulero et al., 2014* | | Produced by Giannakakou Lab |
| Transfected construct | pEGFP-C1-AR-v567 (PC3, M12) | This paper, *Thadani-Mulero et al., 2014* | | Produced by Giannakakou Lab |
| Transfected construct | pEGFP-AR-V7 (PC3,M12) | This paper, *Thadani-Mulero et al., 2014* | | Produced by Giannakakou Lab |
| Transfected construct | pEGFP-C1-AR-fl A573D/ (PC3) | This paper | | Produced by Giannakakou Lab |
| Transfected construct | pEGFP-C1-AR-V7 A573D/ (PC3) | This paper | | Produced by Giannakakou Lab |
| Transfected construct | pEGFP-C1-AR-fl (A596T/S597T) | This paper | | Produced by Giannakakou Lab |
| Transfected construct | pEGFP-C1-AR-V7 (A596T/S597T) | This paper | | Produced by Giannakakou Lab |
| Recombinant DNA reagent | mEos4b-C1 | *Paez-Segala et al., 2015* | | Obtained from Dr. Michael Davidson |
| Transfected construct | mEos4b-AR-fl (PC3) | This paper | | Produced by Giannakakou Lab |
| Transfected construct | mEos4b-AR-v567 (PC3) | This paper | | Produced by Giannakakou Lab |
| Transfected construct | mEos4b-AR-V7 (PC3) | This paper | | Produced by Giannakakou Lab |
| Transfected construct | pCW57.1-GFP-AR-V7 (C42) | This paper | | Produced by Giannakakou Lab |
| Transfected construct | pCW57.1-GFP-AR-V7 DBox (C42) | This paper | | Produced by Giannakakou Lab |
| Transfected construct | pmCherry-C1-RanQ69L (22rv1, PC3) | *Kazgan et al., 2010* | | Obtained from Dr. Jay Brenman |
| Transfected construct | CS-GS241B-mCHER-LV152 lentivirus (PC3) | *Azeem et al., 2017* | | Obtained from Dr. Karl-Henning Kalland |
| Recombinant DNA reagent | pCW57.1 tet-on vector | Addgene | Cat #: 41393 RRID: Addgene_41393 | Obtained from Dr. David Root |
| Software, algorithm | Zeiss Zen Software | Zeiss | RRID:SCR_013672 | https://www.zeiss.com/ microscopy/us/products/microscope-software/zen-lite.html |
| Software, algorithm | GraphPad Prism | GraphPad | RRID:SCR_005375 | http://www.graphpad.com/scientific-software/prism/ |
| Software, algorithm | Fiji | ImageJ | RRID:SCR_002285 | https://imagej.net/Fiji |
| Chemical compound | Fast SYBR Green Mastermix | Thermo Fischer Applied Biosystems | Cat #: 4385612 | |

*Appendix 1 Continued on next page*

*Appendix 1 Continued*

| Reagent type (species) or resource | Designation | Source or reference | Identifiers | Additional information |
|---|---|---|---|---|
| Commercial assay, kit | Macherey-Nagel Kit | Macherey-Nagel | Cat #: 740609.250 | |
| Commercial assay, kit | Nucelofector Kit R | Lonza Bioscience | Cat #: VVCA-1001 | Used LNCaP program T007 with Amaxa II |
| Commercial assay, kit | FuGENE HD Transfection Reagent | Promega Corporation | Cat #: E2691 | |
| Sequence-based reagent | Q-PCR Amplification Primer FKBP5e: Fwd: GGT TCC TGG GCA GGA GTA AG Rev: AAC GTG GAT CCC ACA CTC TC | IDT DNA | | |
| Sequence-based reagent | Q-PCR Amplification Primer PSAe: Fwd-TGG GAC AAC TTG CAA ACC TG Rev-GAT CCA GGC TTG CTT ACT GT | IDT DNA | | |
| Sequence-based reagent | Q-PCR Amplification Primer AREneg: Fwd: GCT GAT TCA ATT ACC TCC CAG AA Rev: AGT TTG GGA CAG ACG GGA AA | IDT DNA | | |

