## [Editor Report]

This work performs a careful study of AR-V7, a splice variant of androgen receptor (AR) that lacks the androgen-binding domain, is constitutively active, and is typically expressed as prostate cancers become resistant to anti-androgen therapies. Clinically, there is intense interest in overcoming anti-androgen resistance, and part of this includes understanding differences between AR-V7 and AR, to be able to therapeutically target AR-V7. This manuscript provides a robust analysis of the regulation of nuclear import and the chromatin-binding features of AR-V7 versus AR. The work reveals that AR-V7 exhibits fast nuclear import kinetics in an NLS- and importin-α/β- independent manner, dependent on the dimerization (D-box) domain mediates AR-V7 nuclear import, revealing a new function for this domain versus its role in the full-length AR. The work also shows that AR-V7 employs an unconventional mode of transcription characterized by high intranuclear mobility, with transient and unstable chromatin interactions, likely reflecting a "Hit-and-Run" mechanism. This greatly enlarges the mechanistic understanding of AR-V7 function, and may help with developing new therapeutic agents.

---

## [Decision Letter]

**Decision letter after peer review:**

Thank you for submitting your article "AR-V7 exhibits non-canonical mechanisms of nuclear import and chromatin engagement in Castrate-Resistant Prostate Cancer" for consideration by *eLife*. Your article has been reviewed by 3 peer reviewers, and the evaluation has been overseen by a Reviewing Editor and Erica Golemis as the Senior Editor. The following individual involved in review of your submission has agreed to reveal their identity: Prof. Dr. John T. Isaacs (Reviewer #2).

Essential revisions:

Overall, the reviewers agree that the experiments are carefully performed and the data are convincing, and the study is important. The reviewers concur that there are some experiments that are essential to solidify the findings.

1. The authors show that neither DTX or IPZ affect AR-V7 nuclear localization, suggesting that nuclear import of AR-V7 is independent of microtubules and importin-B. However, the authors do show that WGA, an inhibitor of nucleoporin-mediated transport, results in cytoplasmic sequestration of AR-V7. They suggest in the discussion that a β-like importin family member with the capacity to mediate import for proteins without a classical NLS might be a candidate importer. This should be investigated more thoroughly and some attempt made to nominate the nuclear transporter for AR-V7, as this would further delineate how the regulation of AR-V7 and AR-FL differ, and is directly related to the main thrust of the authors' manuscript.

2. One major issue throughout is that most studies are done by ectopic expression of tagged plasmids in AR-null cells (PC3). It is unclear if these results would hold in situations where AR-V7 and AR-FL are co-expressed. Some of the key experiments – for example Figure 1F, Figure 2, Figure 3A, etc. should be performed in cell lines co-expressing AR-V7 and AR-FL (22RV1, VCaP, and LN95), with and without isoform-specific knockdown of AR-FL/AR-V7. Additionally, C-terminal endogenous tagging of AR-FL and AR-V7 by CRISPR-mediated knockin, followed by a repeating of some of these same experiments, should be considered.

3. The results in Figure 3A suggest that co-expression of AR-V7 and AR-FL increases AR-FL expression in the nucleus under basal conditions. This is intriguing and may be consistent with the findings of Watson et al., PNAS 2010, suggesting that AR-V7 activity requires AR-FL expression. Increased AR-FL nuclear localization under low androgen conditions may also drive castration resistance, and perhaps AR-V7 drives this indirectly by promoting AR-FL nuclear import, rather than directly. Can the authors determine whether the increased nuclear AR-FL upon AR-V7 co-expression is active (by reporter gene activity or the expression of downstream targets)? Does co-expression or AR-V7 and AR-FL in an AR-dependent line drive increased AR-FL nuclear localization and enzalutamide resistance? Although IPZ treatment indicated that FL-AR import still requires importin, it does not rule out the possibility that FL-AR and AR-V7 may dimerize within the nucleus and lead to increased nuclear retention of FL-AR. This can be tested using a D-box mutant of FL-AR co-expressed with AR-V7.

Optional Revisions

1. Figure 1D – is there a specific reason why this experiment was done in M12 cells while most other experiments seem to be done in PC3?

2. Can the authors more clearly label what is the difference between the "AR-V7 with AR-fl" and "AR-fl with AR-V7" conditions in Figure 3B?

3. Figure 4C – to eye, all the replicate dots look identical (but perhaps there is just very little variability between them?). Please double check this. Statistics also needed for this panel. Also would check some additional AR-V7 target genes as delineated in Cato et al., Cancer Cell 2019 (Figure 2). Particularly given that study and authors' data suggesting that AR-V7 is a repressor, would make sense to check the impact on genes that are repressed, rather than activated by AR-V7.

4. The authors could expand the discussion to emphasis their thoughts as to the implication of the differences between AR FL vs. ARv7 driven transcriptional regulation.

5. The authors have determined that RanGTPase activity and the nucleoporin complex are partially required for nuclear import of AR-V7. It might benefit the readers if the authors can briefly describe what other mechanisms exist which can explain these findings.

6. In Figure 4D, the DNA binding mutant (A573D) does not seem to affect AR-V7 nuclear localization, but it was not quantified and not mentioned in the text. It would be helpful to include the quantification data.

7. In figure 5, the authors showed that co-expression of AR-V7 with FL-AR does not affect AR-V7 mobility when the FL-AR is activated by R1881. It would be interesting to see whether AR-V7 mobility is changed in the presence of Apo-AR as a fraction of Apo-AR may dimerize with AR-V7 and activate a subset of genes using mechanisms which are different from FL-AR and AR-V7 alone.

8. Although it has been suggested that fast nuclear mobility of AR appears to be associated with antagonist bound AR, however, it has also been shown recently that ER antagonist/SERDs immobilized ER on chromatin and which may account for the antagonist activities of these compounds (Guan et al., Cell 2019). Furthermore, with respect to GR, it has been shown that the intranuclear mobility of this receptor correlated with the affinity of the ligand with which this receptor binds. Therefore, these other previous findings should be discussed in the context of the current finding.

9. Line 200, please correct this sentence "FRAP analysis of AR-fl identified slow the fluorescence….

---

## [Author Response]

Essential revisions:Overall, the reviewers agree that the experiments are carefully performed and the data are convincing, and the study is important. The reviewers concur that there are some experiments that are essential to solidify the findings.1. The authors show that neither DTX or IPZ affect AR-V7 nuclear localization, suggesting that nuclear import of AR-V7 is independent of microtubules and importin-B. However, the authors do show that WGA, an inhibitor of nucleoporin-mediated transport, results in cytoplasmic sequestration of AR-V7. They suggest in the discussion that a β-like importin family member with the capacity to mediate import for proteins without a classical NLS might be a candidate importer. This should be investigated more thoroughly and some attempt made to nominate the nuclear transporter for AR-V7, as this would further delineate how the regulation of AR-V7 and AR-FL differ, and is directly related to the main thrust of the authors' manuscript.

We took the reviewer’s suggestion to nominate a putative nuclear import receptor for AR-V7.

As the reviewers suggested to “make an attempt to nominate the nuclear transporter of AR-V7” these are the experiments we performed. Given that AR-V7 does not contain a canonical NLS and ubiquitination of the tumor suppressor PTEN, also without canonical NLS, facilitates its nuclear import via importin-11 (IPO11), we tested the involvement of IPO11 in AR-V7 nuclear import. Using a dominant-negative mutant IPO11, which abrogates its nuclear import function, we show no effect on AR-V7 nuclear localization. This data is now shown in Figure 1- supplement 2. In ongoing studies beyond the scope of this manuscript, we plan to systematically test the involvement of individual nuclear import receptors, including non-canonical importins and transportins with knockdown experiments. The discussion is updated to reflect these new findings.

2. One major issue throughout is that most studies are done by ectopic expression of tagged plasmids in AR-null cells (PC3). It is unclear if these results would hold in situations where AR-V7 and AR-FL are co-expressed. Some of the key experiments – for example Figure 1F, Figure 2, Figure 3A, etc. should be performed in cell lines co-expressing AR-V7 and AR-FL (22RV1, VCaP, and LN95), with and without isoform-specific knockdown of AR-FL/AR-V7. Additionally, C-terminal endogenous tagging of AR-FL and AR-V7 by CRISPR-mediated knockin, followed by a repeating of some of these same experiments, should be considered.

To address this point, we have repeated a few key experiments in cell lines co-expressing endogenous AR-V7 and AR-fl. Specifically, we have confirmed the results shown in Figure 2 on the effect of the catalytic mutant RanQ69L on the impaired nuclear localization of AR-V7, in C4-2 cells with endogenous AR-fl stably expressing inducible GFP-AR-V7. In addition, transient transfection of cells endogenously co-expressing AR-fl and AR-V7, such as 22RV1 cells, corroborated the above findings. These data are now shown in Figure 2-supplment 1 and discussed in results and discussion. We have expanded and clarified the methods to reflect this.

To confirm the partial cytoplasmic accumulation of AR-V7-D-Box, observed following microinjection in PC3 cells (Figure 2C-2D, original submission), we generated C4-2 cells with endogenous AR-fl to stably express doxycycline inducible GFP-AR-V7 D-Box mutant. Quantitation of the subcellular localization of the AR-V7 D-Box mutant revealed statistically significant increase in AR-V7’s cytoplasmic localization in the presence of the D-Box mutations, further suggesting that the D-Box domain may be involved in AR-V7 nuclear localization. These data are now displayed in new Figure 2E-2F and discussed in the text.

Finally, we considered the CRISPR knock-in tagging suggestion, however, it is unfortunately not feasible as AR-V7 is an RNA splice variant and not present in the genome.

3. The results in Figure 3A suggest that co-expression of AR-V7 and AR-FL increases AR-FL expression in the nucleus under basal conditions. This is intriguing and may be consistent with the findings of Watson et al., PNAS 2010, suggesting that AR-V7 activity requires AR-FL expression. Increased AR-FL nuclear localization under low androgen conditions may also drive castration resistance, and perhaps AR-V7 drives this indirectly by promoting AR-FL nuclear import, rather than directly. Can the authors determine whether the increased nuclear AR-FL upon AR-V7 co-expression is active (by reporter gene activity or the expression of downstream targets)? Does co-expression or AR-V7 and AR-FL in an AR-dependent line drive increased AR-FL nuclear localization and enzalutamide resistance? Although IPZ treatment indicated that FL-AR import still requires importin, it does not rule out the possibility that FL-AR and AR-V7 may dimerize within the nucleus and lead to increased nuclear retention of FL-AR. This can be tested using a D-box mutant of FL-AR co-expressed with AR-V7.

As the reviewer suggested, we tried very hard to determine whether nuclear AR-fl in response to AR-V7 co-expression is transcriptionally active. However, due to the very small number of cells co-expressing both constructs, we could not reliably use reporter gene activity which reflects population averaging. To answer this important question, we are currently developing a single-cell sensitive AR-fl-specific fluorescent reporter assay so that we can image single cells that co-express both proteins and get a fluorescent readout, indicative of active AR-fl.

To answer the question of whether co-expression or AR-V7 and AR-FL in an AR-dependent line drive increased AR-FL nuclear localization, we have co-transfected an AR-dependent line (LNCaP) with AR-fl and AR-V7 and observed increased nuclear localization of AR-fl in the presence of AR-V7. In addition, a similar experiment performed in C4-2 cells with inducible AR-V7 confirmed the effect of AR-V7 on enhanced AR-fl nuclear localization. These data are now shown in figure 3C and 3F, respectively. As the number of LNCaP cells co-expressing both AR-Fl and AR-V7 was less than 10% of the transfected population, enzalutamide resistance could not be assessed using population-averaging classical cytotoxicity assays.

Optional Revisions1. Figure 1D – is there a specific reason why this experiment was done in M12 cells while most other experiments seem to be done in PC3?

Yes, the AR-independent M12 cells were stably expressing each construct but they were challenging to culture or microinject/transfect. Therefore, we switched to PC3 cells, which are easily cultured and are a more relevant model.

2. Can the authors more clearly label what is the difference between the "AR-V7 with AR-fl" and "AR-fl with AR-V7" conditions in Figure 3B?

We have changed labeling and legend in the main figure and supplement to highlight which proteins are microinjected in each condition.

3. Figure 4C – to eye, all the replicate dots look identical (but perhaps there is just very little variability between them?). Please double check this. Statistics also needed for this panel. Also would check some additional AR-V7 target genes as delineated in Cato et al., Cancer Cell 2019 (Figure 2). Particularly given that study and authors’ data suggesting that AR-V7 is a repressor, would make sense to check the impact on genes that are repressed, rather than activated by AR-V7.

The reviewer is right that the variation in the values was too small to show in the graph.

4. The authors could expand the discussion to emphasis their thoughts as to the implication of the differences between AR FL vs. aRv7 driven transcriptional regulation.

Yes, we have added elaborating ideas to the Discussion section. We highlight those alternate modes of transcriptional action for AR-V7 may be promoted by differences in protein structure and post-translational modifications relative to AR-fl. Cato et al. suggest that AR-V7 binds preferentially to transcriptional co-repressors, likely due to differences in H3K27 acetylation. How structural differences may impact co-regulator binding with AR-V7 relative to AR-fl remain to be investigated.

5. The authors have determined that RanGTPase activity and the nucleoporin complex are partially required for nuclear import of AR-V7. It might benefit the readers if the authors can briefly describe what other mechanisms exist which can explain these findings.

Our work revealed the partial requirement for RanGTPase activity in AR-V7 nuclear import by utilizing a RanQ96L mutant, which has lowered intrinsic GTPase activity. This suggests that RanGTPase is not the only factor that regulates the nucleo-cytoplasmic transport of AR-V7, yet it plays a role. Functionally, this may suggest that AR-V7 can enter the nucleus even if its nuclear importer is not re-exported into the cytosol. To clarify, our work also showed that AR-V7 import depends completely on its passage through the nuclear pores. We have edited the discussion accordingly.

6. In Figure 4D, the DNA binding mutant (A573D) does not seem to affect AR-V7 nuclear localization, but it was not quantified and not mentioned in the text. It would be helpful to include the quantification data.

The reviewer is correct, we did not observe that the A573D mutation affected AR-V7 nuclear localization based on the quantification and did not include it in the manuscript.

7. In figure 5, the authors showed that co-expression of AR-V7 with FL-AR does not affect AR-V7 mobility when the FL-AR is activated by R1881. It would be interesting to see whether AR-V7 mobility is changed in the presence of Apo-AR as a fraction of Apo-AR may dimerize with AR-V7 and activate a subset of genes using mechanisms which are different from FL-AR and AR-V7 alone.

The reviewers are correct, and that FRAP studies in cells co-expressing of AR-V7 and Apo-AR could potentially suggest a heterodimerization in the nucleus. However, this would be challenging to measure since the fraction of Apo-AR present in the nucleus is likely to be low relative to the amount of AR-V7, and we would not see a meaningful difference in FRAP studies.

8. Although it has been suggested that fast nuclear mobility of AR appears to be associated with antagonist bound AR, however, it has also been shown recently that ER antagonist/SERDs immobilized ER on chromatin and which may account for the antagonist activities of these compounds (Guan et al., Cell 2019). Furthermore, with respect to GR, it has been shown that the intranuclear mobility of this receptor correlated with the affinity of the ligand with which this receptor binds. Therefore, these other previous findings should be discussed in the context of the current finding.

We have modified the discussion to address this point.

9. Line 200, please correct this sentence "FRAP analysis of AR-fl identified slow the fluorescence….

We have corrected this sentence in the manuscript.